# Machine learning-enabled exploration of the electrochemical stability of real-scale metallic nanoparticles

Kihoon Bang[1,2], Doosun Hong[1], Youngtae Park[1], Donghun Kim [2] ✉,
Sang Soo Han [2] ✉ & Hyuck Mo Lee [1] ✉

Surface Pourbaix diagrams are critical to understanding the stability of nanomaterials in electrochemical environments. Their construction based on density functional theory is, however, prohibitively expensive for real-scale systems, such as several nanometer-size nanoparticles (NPs). Herein, with the aim of accelerating the accurate prediction of adsorption energies, we developed a bond-type embedded crystal graph convolutional neural network (BE-CGCNN) model in which four bonding types were treated differently. Owing to the enhanced accuracy of the bond-type embedding approach, we demonstrate the construction of reliable Pourbaix diagrams for very large-size NPs involving up to 6525 atoms (approximately 4.8 nm in diameter), which enables the exploration of electrochemical stability over various NP sizes and shapes. BE-CGCNN-based Pourbaix diagrams well reproduce the experimental observations with increasing NP size. This work suggests a method for accelerated Pourbaix diagram construction for real-scale and arbitrarily shaped NPs, which would significantly open up an avenue for electrochemical stability studies.

When nanomaterials are exposed to external environments such as electric potentials or pH, their surfaces exhibit various phases because the dominant adsorbate species may change[1-4]. The surface phases of nanomaterials can significantly affect the functional properties in energy storage[5,6], sensing[7,8], and catalysis[1,2,9,10] applications. For example, in catalysis, transition metals such as Ag and Ni can be active components in catalysts for oxygen reduction reactions in alkaline conditions and have thus been widely used for alkaline fuel cells[11,12]. However, these elements cannot be utilized in acidic conditions due to their undermined surface stability in acidic pH conditions. Moreover, the catalytic reaction energetics are substantially affected by the surface structure[13,14]. For example, in the water-splitting reaction, the surface oxygen coverage is well known to alter the overall reaction energetics and even change the rate-determining steps[15]. In these regards, to accurately simulate nanomaterials, modeling proper and realistic surface structures under given external conditions is critical.

In electrocatalysis, a surface Pourbaix diagram is the most popular tool to explore the surface structure and stability of catalytic materials[2,16-18] since it reveals the stable surface phases under each applied potential and pH condition. Estimating the surface stability based on Pourbaix diagrams has been widely used for various materials, including pure metals, oxides[19], carbides[20], and even nanoparticles (NPs)[3,21]. In addition to stability evaluations, Pourbaix diagrams can be utilized to explore the adsorbate configurations under certain reaction conditions in various catalysis models[19,22]. Today, the computational construction of surface Pourbaix diagrams is typically based on Gibbs free energy computations at the density functional theory (DFT) level of several possible surfaces phases[2,3,23-25]. Unfortunately, this computational process is prohibitively expensive, as it requires numerous DFT calculations for a wide range of adsorbates and surface coverages, and has thus been limitedly applied to relatively small-size NPs (mostly less than 100 noble metal atoms)[26].

[1]Department of Materials Science and Engineering, Korea Advanced Institute of Science and Technology (KAIST), Daejeon 34141, Republic of Korea. [2]Computational Science Research Center, Korea Institute of Science and Technology (KIST), Seoul 02792, Republic of Korea. ✉e-mail: donghun@kist.re.kr; sangsoo@kist.re.kr; hmlee@kaist.ac.kr

Building Pourbaix diagrams for several nanometer-size NPs involving at least thousands of atoms was considered practically impossible within the current DFT computation frame and speed. This problem leaves the large gaps between experiments and computations unresolved and limits our fundamental understanding of the electrochemical stability of real-scale NPs.

To overcome this problem, machine learning (ML) is a useful tool. After building the database, training, and prediction could be done with personal computers and the computation time is much faster than those of quantum calculations. Not only a speed, but also an accuracy can be achieved with substantial amount of training set[27,28]. Many ML frameworks are used in various materials science fields to predict material's properties from given structures, such as Random Forest Regression[29-31], Gaussian Process Regression[30], and XGBoost Regression[32]. Among them, Crystal Graph Convolutional Neural Network (CGCNN)[33] has many advantages. At first, it can be applied to any kind of material structures by constructing a graph from atomic coordinates, even to NP structures[34]. Moreover, by convolution procedure of graph generated from atomic structures, it takes account local atomic interaction between neighbored atoms which directly influence property of materials.

In this work, we leverage a machine learning (ML) approach to enable the construction of surface Pourbaix diagrams for very large-size NPs. To accelerate the prediction of adsorption energies for a wide range of adsorbates and surface coverages, we develop a bond-type embedded crystal graph convolutional neural network (BE-CGCNN) in which four bonding types (metallic bond, covalent bond, chemisorption, and nonbonded interaction) are uniquely differentiated. BE-CGCNN predicts the adsorption energies for various surface coverages much more accurately than the original CGCNN. Our unique treatment of bond vectors is key to improving ML prediction accuracy and producing reliable Pourbaix diagrams. Using this model, the Pourbaix diagrams for Pt NPs under the competition between O and OH adsorption are accurately produced. Explorations of ML-based Pourbaix diagrams of NPs of various sizes and shapes reveal the origins of the experimentally observed trends, such as the relative dominance of O-covered phases over OH-covered phases for larger Pt NPs and reduction of the Pt ion dissolution area for larger Pt NPs. Owing to the accurate and fast ML predictions, we finally present the construction of Pourbaix diagrams of several nanometer-size NPs involving up to 6525 Pt atoms (~4.8 nm in diameter), which is considered impossible to obtain by DFT only. These demonstrations highlight an ML-enabled tool for exploring the electrochemical stability of real-scale and arbitrarily shaped NPs, which would substantially narrow the gaps between experiments and computations.

## Results

### Bond-type embedded CGCNN (BE-CGCNN)

Previous studies using the ML approach to predict surface adsorption energies mostly focused on single adsorbates[32,35-41]. However, they performed poorly in predicting adsorption energies for cases involving multiple adsorbates and were thus not successful in reflecting surface coverage effects. In this regard, an ML model that can cover various adsorbates and coverages must be developed. For high surface coverage cases, adsorbates on NP surfaces are close enough that several types of interactions can occur, including intermolecular and intramolecular interactions. To reflect such complexity, we propose the bond-type embedded CGCNN (BE-CGCNN) model in Fig. 1, in which edge vectors are uniquely designed.

BE-CGCNN overall follows the schemes of the original CGCNN approach, in particular, those for graph construction, node vectorization, and selection of convolution functions. For each NP and slab structure, atoms and bonds are encoded into node vectors and edge vectors to construct a graph of the corresponding structure. As shown in Fig. 1b, for edge vectors (representing bonds), we classify them into four types: covalent bond within an adsorbate (e.g., O-H), metallic bond within an NP (e.g., Pt-Pt), chemisorption between an NP and an adsorbate (e.g., Pt-O), and, lastly, nonbonded interaction between different adsorbates (e.g., H...O), in which the edge vectors are encoded in a one-hot manner with four categorized vectors. The last term (nonbonded interaction) is only valid when the atomic distance is larger than 1.25 Å. In the previous CGCNN and its modified versions,

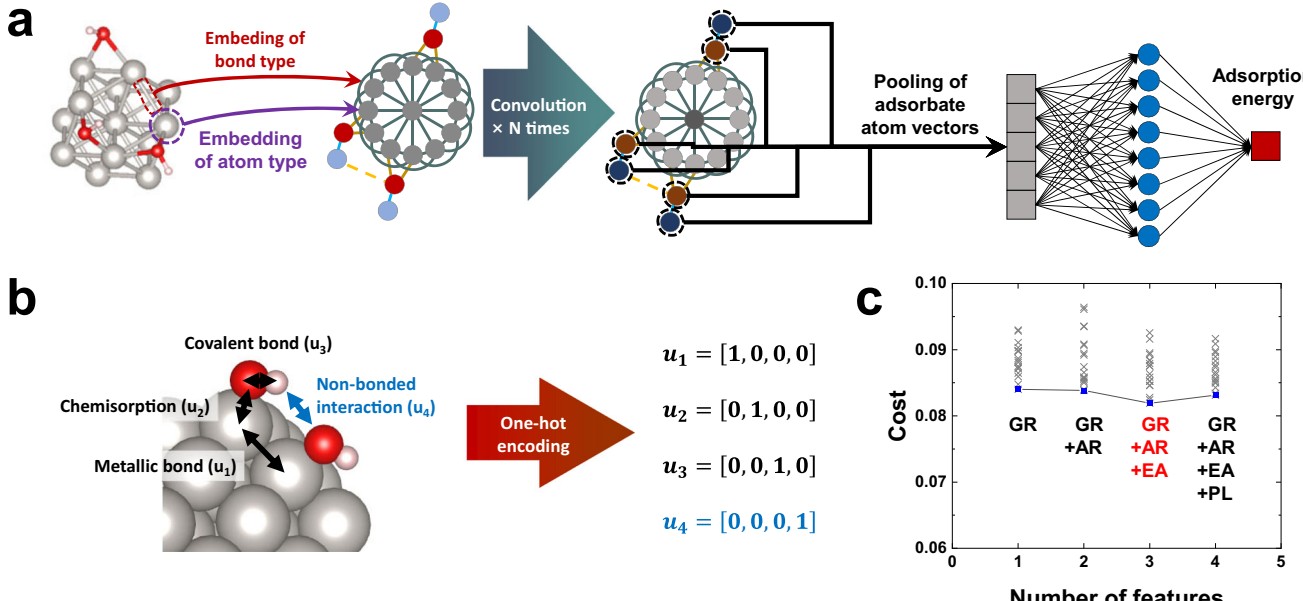

**Fig. 1 | Description of the bond-type embedded CGCNN (BE-CGCNN) model.** **a** Schematic representation of the graph convolution neural network model to predict the adsorption energy. **b** Representation of bond embedding. Each bond is embedded into a bond vector by one-hot encoding of the bond type.

**c** Optimization of atom embedding features. Costs are compared as a function of various feature combinations. Blue points/line denote the minimum value of each feature combination. Here, GR, AR, EA, and PL indicate the group number, atomic radius, electron affinity, and polarizability, respectively.

long-distance interactions were not treated fairly. However, in many cases, there could be strong nonbonded interactions, such as hydrogen bonds, between OH species, and these were treated as important in our graph constructions. Note that the edge vectors are designed to be distance-insensitive.

The processes of node vector construction and optimization are basically identical to those of the original CGCNN. To select appropriate features for the node vector, we calculated the mean square error of the adsorption energy with an increasing number of features, as shown in Fig. 1c. Candidates for the features include elemental properties available in the periodic table of elements as follows: group number, period number, atomic number, radius, electronegativity, ionization energy, electron affinity, volume, atomic weight, melting temperature, boiling temperature, density, $Z_{eff}$, polarizability, resistivity, capacity, number of valence electrons, and number of $d$-electrons. The value ranges and categories are provided in Supplementary Table S1. For cost efficiency, the best feature combination set was fixed with an increasing number of features. The best result was obtained for the feature set comprising group number, atomic radius, and electron affinity. Because the adsorption of adsorbates is closely related to the electronic interaction between adsorbates and NPs, the model with selected features related to the number of valence electrons (group number and atomic radius) and the energy of valence electrons (electron affinity) likely shows the best performance.

## Adsorption energy dataset for various NP surface coverages

In this work, we aim to build surface Pourbaix diagrams of Pt NPs, which are well-known materials in various catalytic applications. In constructing surface Pourbaix diagram, a dataset of adsorption energies on catalyst surface is required. There are several databases containing adsorption energy data in materials science and chemistry field such as OC2020[42] and OC2022[43], but they did not consider enough the coverage of adsorbates and structures of NP-adsorbates, which are critical for Pourbaix diagram constructions. Therefore, we developed our own database by DFT calculations. The adsorption of O and OH species on Pt NPs was calculated for training set construction. Adsorption of O and OH on $Pt_{13}$ and $Pt_{55}$ NPs was included. Two NP structures, cuboctahedron (Coh) and icosahedron (Ih), which are known as the stable morphologies of NPs with 13 and 55 atoms, were considered. NPs with a truncated octahedron (Toh) structure with 38 atoms were also included. Slabs with (100), (110), (111), and (211) exposed surfaces were also included. The structures of these Pt NPs are available in Supplementary Fig. S1. For each NP and slab structure, adsorption configurations of O and OH up to 1 monolayer (ML) were modeled. Here, 1 ML means that adsorbates are fully covered without interacting between neighbored adsorbates. The number of adsorbates for 1 ML for each structure are provided in Table S3 in Supplementary Information. OH was adsorbed on either bridge or top sites and O was adsorbed on bridge or fcc hollow sites, which were the most stable adsorption sites on the Pt(111) slab. For each coverage below 1 ML, up to 5 random configurations where adsorbates are distributed randomly including vertex, edge, and terrace site were modeled to address the effect of configuration to adsorption energies. Configurations with O adsorption-only or OH adsorption-only were considered to reduce the complexity of dataset. The total adsorption energy ($\triangle E[NP - (A_{ads})_n]$) and the adsorption energy per adsorbate ($\triangle E_{ads}[NP - (A_{ads})_n]$) were computed by the following equations:

$$\triangle E[NP - (A_{ads})_n] = E\left[NP - (A_{ads})_n\right] - E[NP] - nE[A] \quad (1)$$

$$\triangle E_{ads}[NP - (A_{ads})_n] = \frac{\triangle E[NP - (A_{ads})_n]}{n} \quad (2)$$

where $E[NP\text{-}(A_{ads})_n]$, $E[NP]$, and $E[A]$ are the total energies of the structures of the NP including n adsorbates, the NP only, and adsorbate A only, respectively. A refers to an adsorbate, such as O and OH.

Unfortunately, neither the total adsorption energy ($\triangle E[NP - (A_{ads})_n]$) nor the adsorption energy per adsorbate ($\triangle E_{ads}[NP - (A_{ads})_n]$) is suitable for accurate ML training for the following reasons. For the former (the total adsorption energy), the data range was estimated to be too large (>120 eV; Supplementary Figs. S2c and S2d) because of the cases involving several tens of adsorbates, and thus, the absolute errors of the ML models are also very large. On the other hand, for the latter case (the adsorption energy per adsorbate), the data range is much smaller (<3 eV; Supplementary Figs. S2a and S2b), and the ML errors seemingly look small. However, in the end, surface Pourbaix diagrams are fed with total adsorption energy inputs, and thus, the predicted values for many adsorbate cases (relatively large surface coverage) would be very erroneous and misleading. To overcome this limitation, we introduced a different metric, namely, the adsorption energy difference ($\triangle\triangle E[NP - (A_{ads})_n]$), which served as a much more suitable form for accurate ML training and prediction and could be computed as follows:

$$\triangle\triangle E\left[NP - (A_{ads})_n\right] = \triangle E\left[NP - (A_{ads})_n\right] - n\overline{\triangle E_{ads}[NP - (A_{ads})_n]} \quad (3)$$

$$\overline{\triangle E_{ads}[NP - (A_{ads})_n]} = \frac{\sum^M \triangle E_{ads}\left[NP - (A_{ads})_n\right]}{M} \quad (4)$$

where $\triangle\triangle E[NP\text{-}(A_{ads})_n]$ is the adsorption energy difference, $\overline{\triangle E_{ads}[NP - (A_{ads})_n]}$ is an averaged value of adsorption energies per adsorbate at each n, and M is the number of adsorption energy data for each n and each adsorbate, and these M values are provided in Supplementary Table S2. The value range of this metric ($\triangle\triangle E[NP-(A_{ads})_n]$) is much smaller (~25 eV; Supplementary Figs. S2e and S2f) than that of the total adsorption energy (>120 eV); thus, the ML errors were also estimated to be much smaller.

## BE-CGCNN training results with bond-type embedding

In Fig. 2, we show the BE-CGCNN model training results obtained using the dataset comprising 736 adsorption energy difference data points (both O adsorption case and OH adsorption case), in which 80% of the dataset was used for training and the remaining 20% was used for the test. For both adsorbates, the BE-CGCNN with bond-type embedding greatly outperforms the original CGCNN, exhibiting a mean absolute error (MAE) of 0.33 eV for the O adsorbate and an MAE of 0.07 eV for OH. These values are much smaller than those of 0.86 eV and 0.49 eV for the O and OH cases from the original CGCNN model ('without bond-type embedding' results in Fig. 2), proving the effectiveness of bond-type embedding in our ML models. Note that these MAE values are remarkably small given that the data range of the adsorption energy difference ($\triangle\triangle E[NP\text{-}(A_{ads})_n]$) is as large as ~25 eV. The error values of the original CGCNN are not sufficiently small to warrant the reliable construction of Pourbaix diagrams, and thus, implementation of BE-CGCNN with much-enhanced accuracy is required.

Different bond-type embeddings lead to different error levels, as shown in Table 1. For each O adsorption and OH adsorption on Pt NPs, the original CGCNN without bond-type differentiation exhibits the worst accuracy. Upon the introduction of three bond types (covalent bond, metallic bond, and chemisorption) into the bond vector encoding, the error is significantly reduced to 0.44 eV and 0.30 eV for each O and OH dataset. Finally, by adding a nonbonded interaction term to the bond types, the ML model reaches even lower-level errors of 0.33 eV and 0.07 eV, respectively. These successful error reductions indicate that the nonbonded interaction is a critical term and should be treated as important, particularly for high surface coverage cases where van der Waals interactions are present.

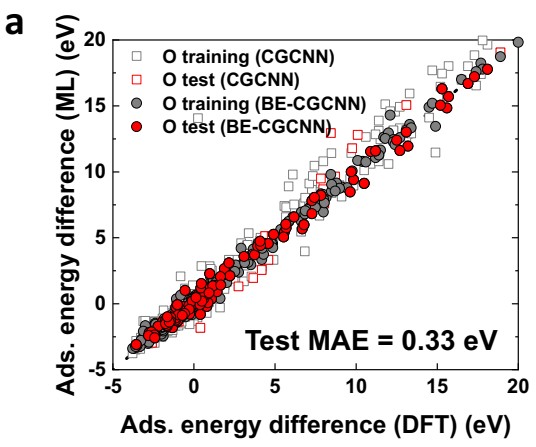 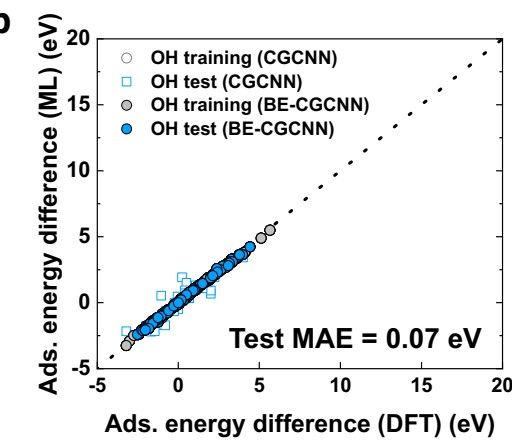

**Fig. 2 | Prediction results of adsorption energy difference by BE-CGCNN model in comparison to those of original crystal graph convolution neural network (CGCNN) model. a** Training and prediction on O adsorbate dataset. **b** Training and prediction on OH adsorbate dataset.

## Constructing Pt surface Pourbaix diagrams using BE-CGCNN

Now, using the well-trained BE-CGCNN model, we are ready to build surface Pourbaix diagrams without DFT inputs. Our model systems are Pt NPs whose surfaces are under the competition between O and OH adsorption. The first validation process is to construct reliable Pourbaix diagrams of $Pt_{55}$(Ih) NPs using the BE-CGCNN model trained on the data of smaller systems, including slabs and smaller NPs ($Pt_{13}$ and $Pt_{38}$). In Fig. 3, the surface Pourbaix diagram of a $Pt_{55}$ NP (Ih) constructed solely by the BE-CGCNN model is compared to the ground truth diagram obtained by DFT computations. The DFT and ML results are observed to be very similar, exhibiting differences in the $y$-intercepts of boundary lines of even less than 0.1 eV on average. In both diagrams, as the pH and U (applied potential) increase, transitions from bare Pt NPs to OH-covered Pt NPs and finally to O-covered Pt NPs are observed. In addition, Pt dissolution regions are found at low pH (->3) and high U (->0.7 V). This test reveals the feasibility of reliable construction of Pourbaix diagrams of larger-size Pt NPs solely based on BE-CGCNN.

Next, we expand the study of ML-based Pourbaix diagrams to Pt NPs of various sizes ($Pt_{55}$ and $Pt_{147}$ NPs) or shapes (Coh and Ih), as shown in Fig. 3. Here, the BE-CGCNN model was trained on the dataset of slabs and relatively small NPs ($Pt_{13}$, $Pt_{38}$, and $Pt_{55}$). In Figs. 3a–d, the surface Pourbaix diagrams of the $Pt_{55}$ (Ih) NP and $Pt_{55}$ (Coh) NP are shown. The diameter of $Pt_{55}$ NPs is ~0.5 nm. Qualitatively, as the pH and given potential U increase, the phase transition from bare Pt to the OH- and O-adsorbed surface occurs. The DFT-based and ML-based predictions appear quantitatively very similar for both shapes. For example, in the Ih shape cases, the phase boundaries between the bare Pt NP and Pt-$(OH)_{0.29ML}$ appear at almost identical $y$-intercepts (0.52 V for DFT versus 0.51 V for ML). Other boundary lines within comparative diagrams of Ih cases appear at very similar positions, with the differences in terms of the $y$-intercept positions being much <0.1 V on average. The diagrams for Coh NP cases are more complicated due to

the appearance of many more phases. Similar to Ih cases, the ML-based predictions overall also worked great for Coh cases, except for the quantitatively erroneous description of the boundary line between Pt-$(O)_{0.67ML}$ and Pt-$(O)_{1ML}$ ($y$ intercept difference of -0.24 V).

We further explore larger-size NP systems, namely, $Pt_{147}$ (Ih) and $Pt_{147}$ (Coh), as shown in Fig. 3e, f. The diameter of the $Pt_{147}$ NPs is ~0.8 nm. For both Ih- and Coh-shaped NPs, Pt-(OH) phases are substantially destabilized over other phases of bare Pt and Pt-(O)[44] in comparison to diagrams of smaller Pt NPs (e.g., $Pt_{55}$) or Pt slab systems[2] available in Supplementary Figs. S4 and S5. This observation is consistent with the tendency of relatively strong OH adsorption on Pt surfaces for smaller Pt NPs compared to bulkier NPs[45].

A prominent difference between $Pt_{147}$ (Ih) and $Pt_{147}$ (Coh) is the relative amount of the fully O-covered phase (Pt-$(O)_{1ML}$). The Pt-$(O)_{1ML}$ phase stands out for the Ih case, whereas this phase shrinks for the Coh structures due to the appearance of partial O coverage phases, such as Pt-$(O)_{0.53ML}$. Reduction of the Pt-$(O)_{1ML}$ phase was also similarly found for smaller $Pt_{55}$ NPs (Coh), where partial O coverage phases (Pt-$(O)_{0.42ML}$, Pt-$(O)_{0.56ML}$, Pt-$(O)_{0.67ML}$) preferentially appear over the full O coverage phase. This difference comes from the shape effect of the NPs. Unlike the Ih structures comprising only (111) surface planes, Coh NP structures have mixed surfaces of (111) and (100) planes. The adsorption of O species on the Pt(100) surface is well known to be stronger than that on the (111) surface due to the presence of more dangling bonds[46]. As a result, for the Coh NP structures, partial O phases (Pt-$(O)_{<1ML}$) are likely to be stable when oxygen is dominantly adsorbed on (100) surfaces compared to the full O phase (Pt-$(O)_{1ML}$) where oxygen is adsorbed on both (100) and (111). This phenomenon would be unlikely for the Ih NP structures without (100) surfaces.

The BE-CGCNN model is not only applicable to highly symmetric NPs such as Ih and Coh, but also to asymmetric NPs which could be also synthesized in experimental condition[47,48]. To validate the performance of BE-CGCNN model for the non-idealized NP shapes, surface Pourbaix diagram of asymmetric NPs were built based on the prediction of adsorption energy difference. For more realistic modeling, the shapes of asymmetric NPs were generated using the heating-and-quenching approach of $Pt_{55}$ NPs in molecular dynamics (MD) simulation of As a result, two different structures of asymmetric $Pt_{55}$ NP structures (AS1, AS2) could be obtained. The details of the structure generation procedures are provided in Supplementary Fig. S6. In Fig. S6, the surface Pourbaix diagrams of $Pt_{55}$(AS1) and $Pt_{55}$(AS2) are shown and compared with DFT-computed diagrams. Because asymmetric NP structures are not included in training set, the difference between BE-CGCNN and DFT-based diagrams are larger than the difference for the case of $Pt_{55}$(Ih) or $Pt_{55}$(Coh). Nevertheless, the same phases appear in

## Table 1 | MAE values with the variations of bond-type embedding

| Bond embedding | Test MAE (O DB) [eV] | Test MAE (OH DB) [eV] |
|---|---|---|
| Without bond-type embedding | 0.86 | 0.49 |
| Bond-type (C, M, CH) | 0.44 | 0.30 |
| Bond-type (C, M, CH, NB) | 0.33 | 0.07 |

*C* covalent bond, *M* metallic bond, *CH* chemisorption, *NB* nonbonded interaction.

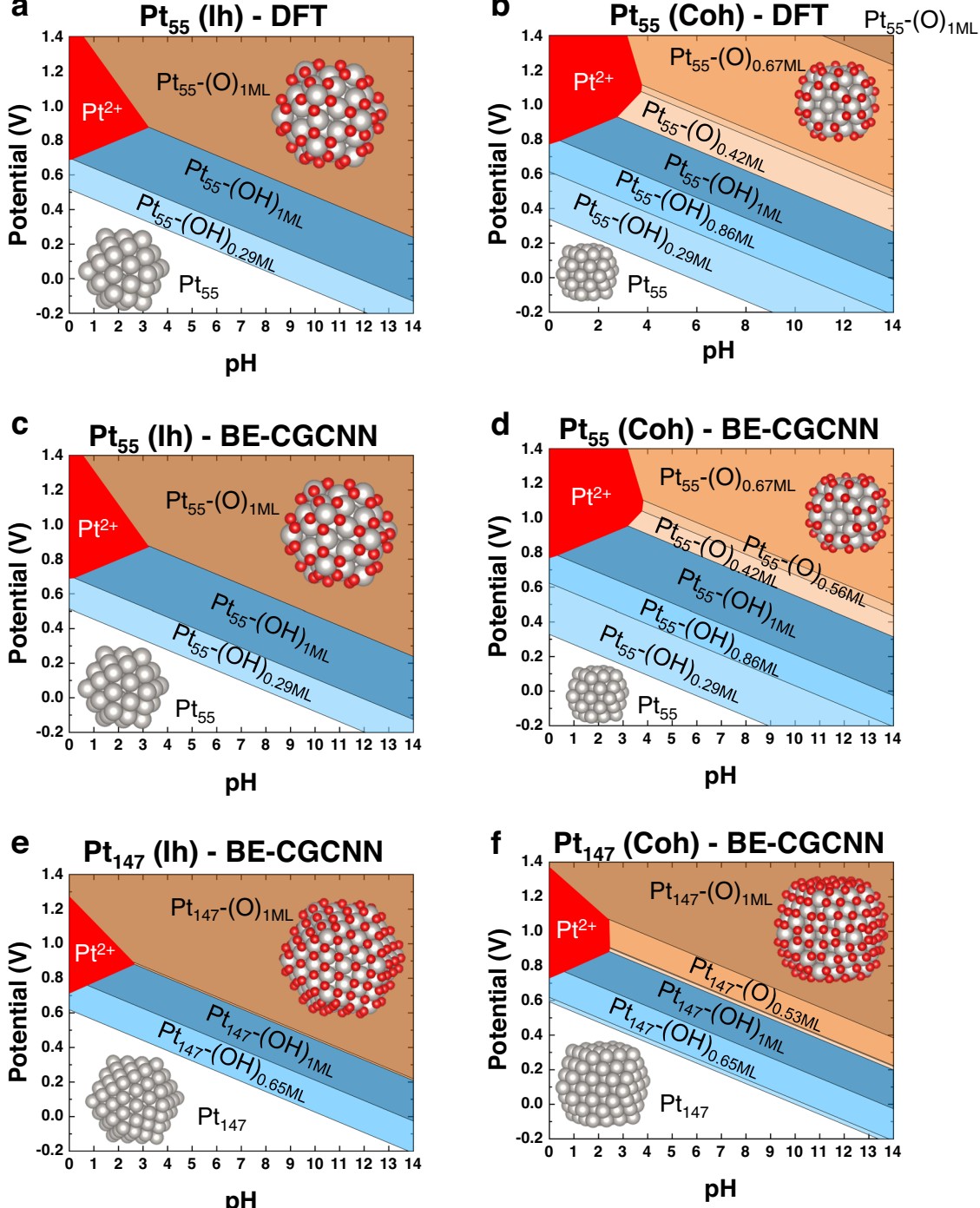

**Fig. 3 | Surface Pourbaix diagrams of various size and shape Pt nanoparticles (NPs) based on density functional theory (DFT) calculation or BE-CGCNN prediction. a** $Pt_{55}$ (Ih) based on DFT. **b** $Pt_{55}$ (Coh) based on DFT. **c** $Pt_{55}$ (Ih) based on BE-CGCNN. **d** $Pt_{55}$ (Coh) based on $Pt_{147}$ (Coh) based on BE-CGCNN. **e** $Pt_{147}$ (Ih) based on BE-CGCNN. **f** $Pt_{147}$ (Coh) based on BE-CGCNN. The white, blue, orange, and red shaded areas represent bare Pt NPs, OH-covered, O-covered, and Pt dissolution phases, respectively. As the color became darker, more adsorbates are adsorbed. The red and gray spheres of the inset atomic models represent O and Pt atoms, respectively.

both diagrams and the trend is very similar, which proves the effectiveness of BE-CGCNN for non-idealized NP shapes.

Interestingly, the shape of diagrams of asymmetric NPs are quite different from that of $Pt_{55}$(Ih) despite the same size. The main difference is that the area of bare Pt phase is expanded for AS1 and AS2 NPs. As the asymmetric NP passes through melting and quenching process, the surface become smoothened, and thus the number of low-coordinated surface atoms such as vertex and edge (typically stronger binding sites) become lower. Therefore, the average adsorption energy per adsorbate become weaker from −2.90 eV for OH and −4.18 eV for O on Ih and Coh NPs to −2.35 eV for OH and −3.61 eV for O on AS1 and AS2 NPs. This result adequately explains the expanded bare Pt region in Pourbaix diagrams for asymmetric NPs.

Although we have thus far shown that BE-CGCNN model can produce surface Pourbaix diagrams with fairly high accuracy, it does not tell whether or not the ML predicted results are trustworthy. To estimate

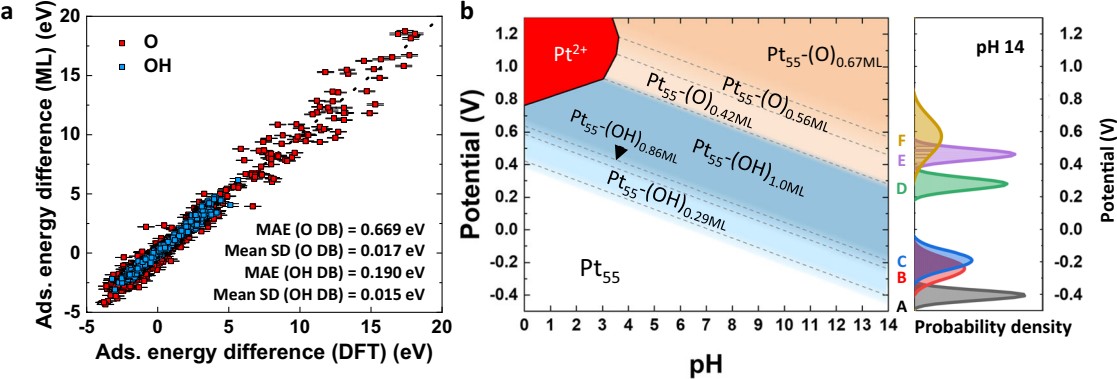

**Fig. 4 | Uncertainty estimation and its reflection to surface Pourbaix diagram.**
**a** Machine learning (ML) (BE-CGCNN with Dropout Neural Network in this case) and DFT prediction results of adsorption energy difference. Each point represents average value of 1000 sampled case of predicted adsorption energy difference and the corresponding error bar represents ± one standard deviation value. **b** Surface Pourbaix diagram of $Pt_{55}$ (Coh) with uncertainty estimations. The phase boundary line is represented by the dashed line (the average of 1000 sampled cases), and its uncertainty is represented by the gradient at each line. For a clearer understanding of the uncertainty values, the probability density distributions of each phase boundary lines at pH 14 are shown on the right. The white, blue, orange, and red shaded area represent bare Pt NPs, OH-covered, O-covered, and Pt dissolution phases, respectively. As the color became darker, more adsorbates are adsorbed.

the model's reliability, uncertainty quantification could be a useful tool. By estimating uncertainty values for ML predictions, the model not only provides a prediction value but also returns confidence intervals. There are many available methods to measure uncertainty[49–51], and among them, we adopted Dropout Neural Network (Dropout NN) which is directly applicable to our BE-CGCNN model with ease. It approximates Bayesian models by enforcing dropout at the ML prediction stages. We used a random dropout rate of 25% to each convolution layer before fully connected neural network layers in the BE-CGCNN model. 1000 times of sampling was performed for each prediction, and the uncertainty value can be obtained as a standard deviation (SD) of 1000 times predictions from the committee of ML models.

This model with Dropout NN was trained with each O and OH DB, and prediction result is shown in Fig. 4a. It slightly underperforms the model without Dropout NN, as confirmed by the increased MAE values of 0.67 eV and 0.19 eV for each O and OH adsorbates. On the other hand, the SD values are 0.017 eV and 0.015 eV for each O and OH DB, respectively, which are very small values considering the very wide range of adsorption energy difference in our dataset. This result indicates that BE-CGCNN model is highly reliable. In addition, surface Pourbaix diagram can be built based on the model with Dropout NN, as shown in the example of $Pt_{55}$ (Coh) in Fig. 4b. In the diagram, the main phase boundary lines were determined from the average of the ML prediction values (average of 1000 sampled cases), while the uncertainty of boundary lines were calculated by adding and subtracting of SD values at each line. We observe in Fig. 4b that the Pourbaix diagram of $Pt_{55}$ (Coh) is qualitatively and quantitatively similar to the results obtained without Dropout NN, since the prediction accuracy is not much undermined. The uncertainty range of phase boundary lines would reveal the reliability of the boundary lines. In the case of $Pt_{55}$ (Coh) in Fig. 4b, the largest SD of phase boundary line is only 0.117 V for the phase boundary between $Pt_{55}$-$(O)_{0.56ML}$ and $Pt_{55}$-$(O)_{0.67ML}$, and the other boundary lines are much more confident as confirmed by smaller SD (0.031-0.066 V). The BE-CGCNN model with Dropout NN is highly beneficial for predicting the uncertainty of the predicted phase boundary lines of Pourbaix diagrams. Nonetheless, since the overall trends of surface Pourbaix diagram are not affected much by the inclusion of Dropout NN, the remaining studies were performed with BE-CGCNN without Dropout NN.

## Pourbaix diagrams for real-scale Pt NPs
The NPs explored thus far are NPs involving up to 147 atoms. This size corresponds to only 0.8 nm in diameter, which is unfortunately far

from the real scale. Typically, the diameters of experimentally synthesized NPs are over 3–4 nm, which involves thousands of atoms. Using BE-CGCNN, we demonstrate the construction of Pourbaix diagrams of real-scale Pt NPs, including $Pt_{561}$ (Coh), $Pt_{3871}$ (Coh), and $Pt_{6535}$ (Coh) NPs (~2.8, 3.9, and 4.8 nm in diameter), in Fig. 5a–c. Since these diagrams are for several nanometer-size Pt NPs, we can now compare the results with the available experimental reports. For example, there are studies reporting the measured onset potentials for surface oxide (or fully O-covered phase) generation on Pt NPs, including 0.96 V on 1.2 nm-size NPs from Merte et al.[52] and 0.9–1.15 V on 4.0 nm-size NPs from Mom et al.[53] These values are marked on the Pourbaix diagrams in Fig. 5a, b and are found near the boundary lines of Pt-$(O)_{1ML}$ in each diagram. Such quantitative agreement well supports that the constructed Pourbaix diagrams are highly reliable and that the size dependences are greatly reflected.

As the NP size increases, the OH-covered phases are observed to be reduced compared to the O-covered phase. This is consistent with the experimental report that the OH surface coverage decreases with increasing NP size[44]. The Pourbaix diagrams asymptotically converge above the size of 3871 atoms. The converged case can be compared to the Pourbaix diagram of the Pt(111) slab system, available in Supplementary Fig. S5: the compositions of the O-covered phases are quite different, whereas those of the bare Pt phases and OH-covered phases are very similar. This difference arises because the Coh-shaped NPs have (100) terraces, unlike the Pt(111) slabs.

The increasing O- to OH-covered phase ratio with increasing NP size can be understood in terms of the relative adsorption strength of O and OH species. Figure 5d compares the adsorption energies per adsorbate of the fully O-covered (Pt-$(O)_{1ML}$) and OH-covered (Pt-$(OH)_{1ML}$) phases, and their difference becomes larger with increasing NP size, which is consistent with the shrunken OH phases for larger NPs. The NP size dependence of the adsorption energies can be adequately explained by the relative adsorption strength on vertex sites, edge sites, and terrace sites. Taking $Pt_{55}$ (Coh) NPs as an example, the adsorption of OH on a vertex site is 0.68 eV stronger than that on a terrace site, whereas the adsorption of O on a vertex site is only 0.27 eV stronger than that on a terrace site. The smaller the Pt NPs become, the larger the ratio of vertex and edge sites to terrace atom sites becomes, and thus, the adsorption energy difference between O and OH species ($E_{ads, O\ 1ML} - E_{ads, OH\ 1ML}$ in Fig. 5d) is gradually reduced. These trends are clearly presented in Fig. 5d.

In addition to the O- to OH-covered phase ratio, the Pt dissolution phase is also an interesting spot to focus on. As the NP size increases,

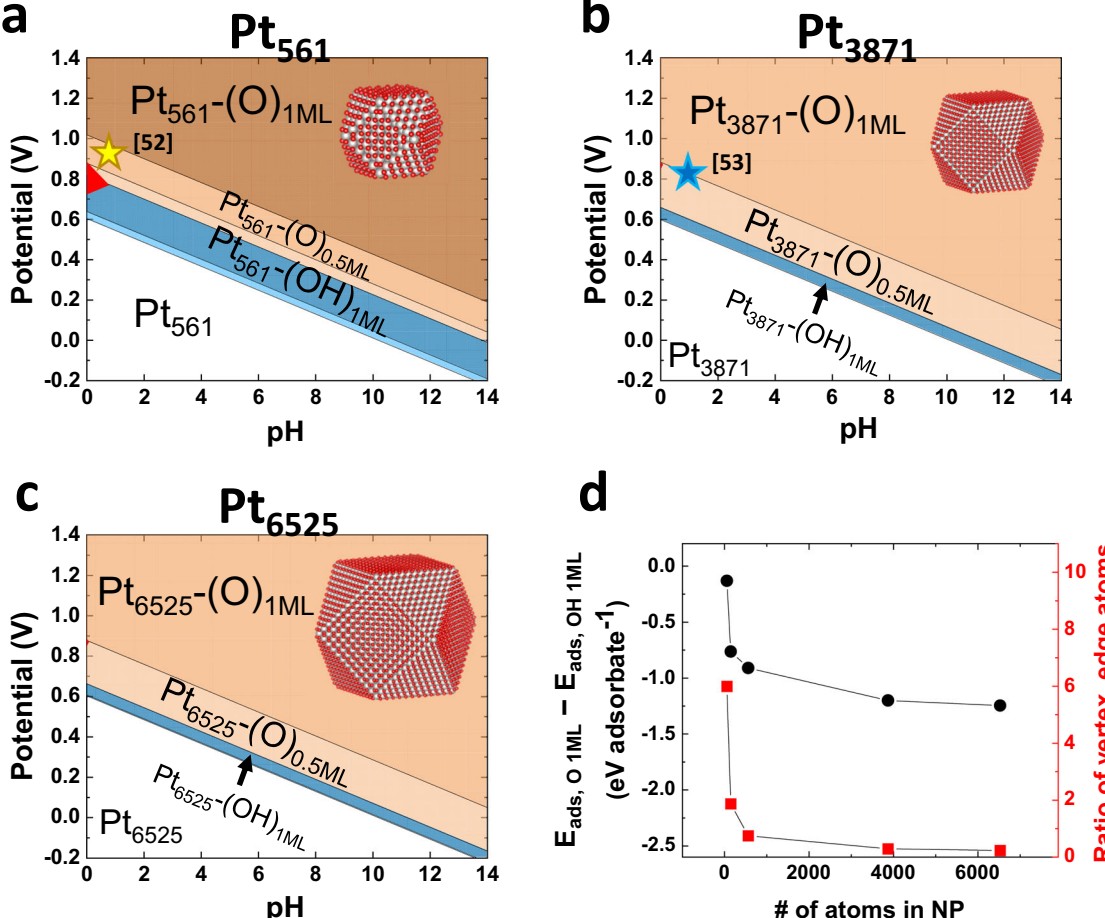

**Fig. 5 | Pourbaix diagrams of real-scale (several nanometer-size) Pt NPs.**
**a** Pourbaix diagram of $Pt_{561}$ (Coh). **b** Pourbaix diagram of $Pt_{3871}$ (Coh). **c** Pourbaix diagram of $Pt_{6525}$ (Coh), which corresponds to ~4.8 nm in diameter. The white, blue, orange, and red shaded area represent bare Pt NPs, OH-covered, O-covered, and Pt dissolution phases, respectively. As the color became darker, more adsorbates are adsorbed. The red and gray spheres of the inset atomic model represent oxygen and platinum atoms, respectively. **d** Difference in the adsorption energies between the fully O-covered phase ($Pt-(O)_{1ML}$) and fully OH-covered phase ($Pt-(OH)_{1ML}$) as a function of NP size. $E_{ads}$ represents the adsorption energy per adsorbate. The 2nd $y$ axis shows the ratio of the number of vertex and edge atoms to the number of terrace atoms as a function of NP size.

the Pt dissolution phases shrink. The Pt dissolution area is closely related to the stability of NPs operating in electrochemical catalysis. For example, Pt NPs are one of the most widely used catalysts for the oxygen reduction reaction (ORR) at the cathode of proton exchange membrane fuel cells (PEMFCs). However, the operating conditions of the PEMFC cathode are approximately an applied potential of 0.8 V and a pH of 1[54]. This point is inside the Pt dissolution domain for relatively small Pt NPs ($Pt_{55}$ and $Pt_{147}$), as shown in Fig. 3, whereas it resides outside of the Pt dissolution region for larger Pt NPs ($Pt_{561}$, $Pt_{3871}$, and $Pt_{6525}$), as shown in Fig. 5a–c. Very interestingly, an experimental report[55] confirmed that Pt NPs with diameters smaller than 2.0 nm were easily dissolved, and thus, the specific electrochemical surface area was greatly decreased over potential cycling compared to larger NPs.

We importantly note that the experimental operating conditions (0.8 V, pH = 1) are near the fully O-covered phase, which is a surface Pt oxide. Although this position is outside of the Pt dissolution region, the surface Pt oxide layer induces nonequilibrium transient dissolution of Pt[56–58]. In these situations, Pt dissolution may occur even for large-size Pt NPs, and the ORR performance of the catalyst would be degraded under long-term working conditions. This indicates that to fully understand the electrochemical stability of NP-based catalysts, taking into account kinetic factors, not only the surface Pourbaix diagram in which only equilibrium states are considered, is also important.

The construction of Pourbaix diagrams of several nanometer-size NPs was enabled by the fast prediction speed of BE-CGCNN compared to DFT. Thus, discussing the computing time taken for the task of predicting the adsorption energy of each structure by the BE-CGCNN and DFT methods should be worthwhile. For $Pt_{147}$ NPs, the total computing time (both training and prediction) for BE-CGCNN was estimated to be ~150 seconds based on a personal computer implemented with an NVIDIA GPU of GTX 2070. In contrast, the computing time for the same task was ~90 hours (2160 times longer than in the BE-CGCNN case) based on a high-performance computing node implemented with a 2.3 GHz 20-core CPU, as shown in Fig. 6. Because DFT theory follows the computational scaling of $O(N^3)$, where N is the number of electrons[59,60], the computing time differences between BE-CGCNN and DFT will be large for NPs involving thousands of atoms. Following the extrapolation lines based on $O(N^3)$ for DFT, for the example of the NP of 6535 Pt atoms, the computing time of DFT will be 2200 days, which is ~$1.9 \times 10^8$ times longer than that in the BE-CGCNN case.

## Discussion
In summary, we solved the problem that the construction of Pourbaix diagrams of real-scale NPs is not practically possible today due to the extreme DFT cost issue. As a first step to solve this problem, we

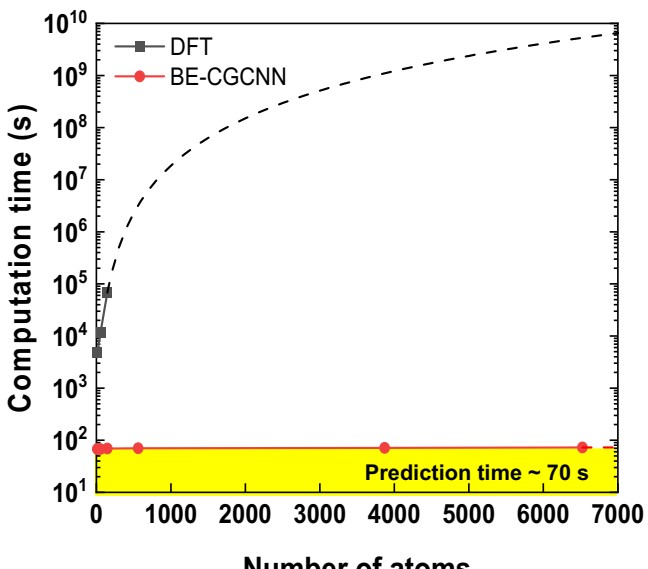

**Fig. 6 | Computational cost of BE-CGCNN model training and predictions, compared to DFT computations.** Dashed line is extrapolated result for NPs larger than 147 atoms. The standard DFT theory follows the scaling of $O(N^3)$ where $N$ denotes the number of electrons. The yellow shaded area represents the prediction time by BE-CGCNN.

developed BE-CGCNN, in which four bonding types were uniquely encoded. BE-CGCNN substantially outperforms the original CGCNN in predicting adsorption energies over a wide range of NP surface coverages. Using BE-CGCNN, we demonstrate the construction of Pourbaix diagrams of Pt NPs involving up to 6535 atoms (~4.8 nm in diameter). Exploring Pt NPs of various sizes and shapes, we find that the ML-based Pourbaix diagrams well reproduce experimental observations, such as an increasing O- to OH-covered phase ratio and a decreasing Pt dissolution area as the NP size increases. By presenting surface Pourbaix diagrams of very large-size NPs, we conclude that BE-CGCNN can serve as a strong tool to enable the stability study of real-scale and arbitrarily shaped NPs in electrochemical environments, which is not possible in the conventional DFT scheme. Currently, our model is limited to specific systems as our dataset is only composed of Pt-O or Pt-OH structures. The model will not function well for other adsorbates (e.g., OOH, CO) or for different composition of NPs (e.g., $Pt_3Ni$, $Pt_3Fe$). However, if we precisely prepare a training set for the system we interested in and follow the similar protocol, BE-CGCNN model would be effective for the expanded material spaces, which remains as a future research.

## Methods

### DFT computation
To calculate the adsorption energies for construction of surface Pourbaix diagram, we performed spin-polarized DFT calculations using the Vienna Ab initio Simulation Package (VASP)[61,62] with the projector-augmented-wave pseudopotentials[63] and the revised Perdew-Burke-Ernzerhoff (RPBE)[64,65] gradient approximation was used for the exchange-correlation functional. To treat van der Waals interactions between adsorbates, Grimme's DFT-D3 method[66] was adopted. Another function of local-density approximation (LDA) was also tested, and its results are compared to the case with RPBE+D3 in Fig. S8. LDA was found to generally overestimate adsorption energies, causing unrealistic Pourbaix diagrams. In contrast, RPBE+D3 is known to produce accurate adsorption energies with reasonable computational cost compared to other functionals[67,68]. The plane-wave cutoff was set to 520 eV, and the convergence criteria for electronic structure and

geometry optimization was $3 \times 10^{-5}$ eV and 0.05 eV/Å. For each NP and NP-adsorbates structures, a vacuum spacing of 10 Å was used to prevent interactions between NPs.

### Gibbs free energy corrections
For Pourbaix diagram construction, Gibbs free energies need to be computed, which involves correction terms of the entropy (ΔS), zero point energy (ΔZPE), applied potential (U), and pH. Within the computational hydrogen electrode scheme, the Gibbs free energies of each NP-adsorbate structure were computed from the following equations. Note that at a certain surface coverage (or a certain number of adsorbates), the lowest adsorption energy ($\triangle E[NP - (A_{ads})_n]$) case was selected for Gibbs free energy computations.

$$\Delta G[NP - (O)_n] = \Delta E[NP - (O)_n] + n(E[O] - (E[H_2O] - E[H_2]) \\ - 2(eU + 0.0592pH) + \Delta ZPE[O] - T\Delta S[O]) \quad (5)$$

$$\Delta G[NP - (OH)_n] = \Delta E[NP - (OH)_n] + n(E[OH] - \left(E[H_2O] - \frac{1}{2}E[H_2]\right) \\ - (eU + 0.0592pH) + \Delta ZPE[OH] - T\Delta S[OH]) \quad (6)$$

where ΔS was approximated from the loss of entropy from the gas phase molecule upon binding to the surface. The NP-adsorbate structures with the lowest (most stable) Gibbs free energy are shown on the surface Pourbaix diagram for given U and pH values.

### Computation of dissolution phases in Pourbaix diagrams
We defined the dissolution phase of a $Pt_n$ NP as when one monolayer of the $Pt_n$ NP shell dissolves into Pt ions. The Gibbs free energy of the dissolution phase ($\Delta G[Pt_{n,diss}]$) can be computed as follows:[69]

$$\Delta G[Pt_{n,diss}] = G[Pt_m] - G[Pt_n] + n_{shell}(G[Pt_{bulk}] - 2e(U - U_{diss,bulk})) \quad (7)$$

where $n_{shell}$ is the number of atoms in one monolayer of the $Pt_n$ NP, m is the number of atoms in dissolved NPs (i.e., $m+n_{shell} = n$), $U_{diss,bulk}$ is the dissolution potential of bulk Pt, and $G[Pt_n]$ is the Gibbs energy of the $Pt_n$ NP.

As shown in the equation, we need the Gibbs free energy of NPs ($G[Pt_n]$) to obtain the Gibbs free energy of the dissolution phase ($\Delta G[Pt_{n,diss}]$). It can be obtained from DFT calculations for small-size NPs; however, the computation cost would be very high for large-size NPs involving hundreds of metallic atoms. To overcome this problem, we calculated the energy of Pt NPs by the classical forcefield. Here, we applied the second nearest-neighbor modified embedded-atom method (2NN-MEAM) of J.–S. Kim et al.[70] As shown in Supplementary Fig. S3, the energies calculated by the 2NN-MEAM forcefield are ~0.3–0.8 eV larger than the DFT calculated energies; however, the trends appear to be very similar. Because the relative energy, not the absolute energy, between NPs is important to calculate the dissolution phase, 2NN-MEAM could be a substitute for DFT. Also, in the work of evaluation of forcefield for Pt[71], MEAM showed quite good performance even for NP structures. Thus, we used the 2NN-MEAM forcefield energy for the computation of the Gibbs free energy of the dissolution phase.

### BE-CGCNN model development
In the BE-CGCNN model, for the slab structure and each NP structure, atoms and bonds within NPs were encoded into node vectors and edge vectors to construct a graph of the corresponding structures. The node vector construction processes are basically the same as those of the original CGCNN. The elemental properties are available in the periodic table of elements were used as candidate features, such as the

group number, period number, and electronegativity. The value ranges and categories are provided in Supplementary Table S1. In contrast, the edge vectors were uniquely designed. Following the previously reported slab graph convolutional neural network (SGCNN) work[72], the edge vectors were intentionally set to be distance-insensitive. As a result, the connectivity information required as an ML input is whether arbitrary atom pairs are connected (yes or no) instead of their distance values. In this scheme, we no longer need fully DFT-relaxed structures as ML inputs. The additional key modification for edge vectors, which is unique in this work, is that four bonding types were treated differently. These four bonding types include covalent bonds within adsorbates (e.g., O-H), metallic bonds within NPs (e.g., Pt-Pt), chemisorption between the NP surface and adsorbates (e.g., Pt-O), and finally nonbonded interaction between neighboring adsorbates (e.g., O···H with a distance of over 1.25 Å). The treatments of these four bonding types in edge vectors are well described in Fig. 1 and related explanations.

The constructed graphs were followed by several convolutional layers. For each convolutional layer, each node (atom vector) was updated based on the following convolution functions:

$$z^t_{(i,j)} = v^t_i \oplus v^t_j \oplus u_{(i,j)} \tag{8}$$

$$v^{t+1}_i = v^t_i + \sum_j \sigma(z^t_{(i,j)} W^t_f + b^t_f) \odot g(z^t_{(i,j)} W^t_s + b^t_s) \tag{9}$$

where vectors $v$ and $u$ are node and edge vectors, subscripts $i$ and $j$ denote neighboring atoms, and superscript $t$ denotes the number of convolutional layers. The operation $\oplus$ denotes concatenation, $\odot$ denotes elementwise multiplication, $\sigma$ denotes the sigmoid function, and g is the rectified linear unit (ReLU) function. The pooling process after convolutions was performed by normalized summation of convolutionized atom vectors. The pooled vector was finally related to the adsorption energy term (the final output) via fully connected neural networks (FCNs). The cost function of model was set to $L^2$ loss with $L^2$ regularization. The optimized hyperparameter results are as follows: 32 batch size, 1000 epochs, a 0.001 learning rate, 5 convolutional layers, and 2 FCN hidden layers with 25 nodes. Dropout[73] and $L^2$ regularization were applied to overcome overfitting. The dropout property and $L^2$ regularization coefficients were 0.3 and 0.01, respectively. We trained the model for 1000 epochs and the parameters where the cost of validation set is the lowest were selected, as shown in Fig. S12.

### Reporting summary
Further information on research design is available in the Nature Portfolio Reporting Summary linked to this article.

## Data availability
Related data are available at https://github.com/kihoon-bang/GCNN_bond_embedding, or from the corresponding authors on request.

## Code availability
The implemented ML model code is available at https://github.com/kihoon-bang/GCNN_bond_embedding[74], or from the corresponding authors on request.

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

## Acknowledgements
This work was supported by the Samsung Research Funding & Incubation Center of Samsung Electronics under Project Number SRFC-MA1801-03.

## Author contributions
K.B. performed the DFT calculations, developed the ML models, analyzed the data, and prepared the manuscript. D.H. assisted to develop the ML models and analyze the data. Y.P. assisted with the DFT calculations. D.K., S.S.H., and H.M.L. designed and directed the studies. All authors contributed to writing and reviewing the manuscript.

## Competing interests
The authors declare no competing interests.
