## [Peer Review File · Nature Communications]

Machine Learning-Enabled Exploration of the Electrochemical Stability of Real-Scale Metallic NanoparticlesReviewers' Comments:

Reviewer #1:

Remarks to the Author:

The work reports on the machine learnt accelerated DFT-based construction of the Pourbaix Diagram of nanoparticles. The topic is of high relevance both for fundamental and applied science motivations. Further, it demonstrates a step forward in embracing a realistic complexity in modelling, so to close the complexity gap between simulations and experiments.

I'm very positive about the work, and would ask few additional clarifications/calculations, so as to improve the impact and reach of the paper, and fully match the quality of reports disseminated in Nature Communications:

-) I think that it is necessary to report on the uncertainty of machine learning models predictors for adsorption energy calculation. This could be readily extracted from, e.g., predictions obtained from committee of models. I refer the authors to Journal of Chemical Physics 154 (7), 074102 2021 for a detailed calculation of unbiased uncertainties given a committee of machine learning models, and to Mach. Learn.: Sci. Technol. 1 025006, 2020 in case they would prefer to adopt alternative uncertainty estimates.

Note, It is also very interesting to assess how uncertainty propagates into the Pourbaix Diagram calculation. I expect that rules of propagation of error would render this rather feasible, and provide an even more grounded estimate of the agreement between the model and experiments.

-) It would be interesting to report on how the Pourbaix Diagram changes depending on the DFT functional choice. While RPBE+D3 is a perfectly reasonable choice to calculate adsorption energies, other properties e.g., thermodynamic stability of a given nanostructure, are better captured by other DFT functionals (see e.g., Nature Communications 12 (1), 1-9 2021).

-) The discussion presented in Adsorption energy dataset for various NP surface coverages should be clarified. In particular "and M is the number of each n adsorbate case in our dataset" is a confusing sentence. yet, it is of key importance to understand the metric introduced by the authors.

-) in the abstract the authors highlight the capability of the model for "arbitrarily shaped" NPs, yet they present a discussion solely on closed-shell nanoparticles with no defects. Can the authors demonstrate the effectiveness of the model for defected/asymmetric shapes ? are the trends they observe confirmed also for non-idealised shapes ?

Note, the authors correctly included less-ideal shape in the training of machine learning models for DOS in nanoparticles (Scientific Reports 11, 11604 (2021)). Also, defected Pt nanoparticles may be very active for, e.g., Oxygen Reduction (see J. Am. Chem. Soc. 2020, 142, 42, 17812-17827 and Acs Catalysis 10 (6), 3911-3920, 2020 just to name a few).

Reviewer #2:

Remarks to the Author:

This manuscript addresses the complexity of predicting surface Pourbaix diagrams for large-size nanoparticles that would ordinarily be outside of the bounds of ab initio DFT. By using an extension to CGCNNs, this machine learning approach accounts for four bonding types explicitly. The reproduction of surface Pourbaix diagrams that were created with DFT shows the utility of this approach and as such would be of impact when considering stability of realistic systems. Primarily the comments included here deal with the reporting of methodological details. I do not think the information currently provided is enough to reproduce results so it would be quite helpful to have more insight into the how the training dataset was defined.

Comments and requested clarifications:

133: The intuition for choosing the feature set seems reasonable, coupled with the figure 1c. However it would be helpful if the cost function were explicitly defined somewhere. The differences in cost seem quite small and I would naively expect that a combination of features that least relate to adsorption energy and valence electrons would result in a higher cost and more lower accuracy.

139: Fig 1b typo in "non-bonded interaction u_4 ". Text in figure 1b is a bit inconvenient to read on top of the atoms. Also why is u_4 shown in red? I thought it was related to the red text in 1c but that is not the case.

157: Would be great to know more details about the kind of adsorption configurations were used in the dataset to construct the CGCNNs. This is a large sample space after all (In 190: "736 adsorption energy difference points). Perhaps partly this information may be added in the SI. For example: How many adsorbates are in 1ML on each NP (and on what sites)? Were mixed phases considered between O and OH adsorbates (line 157 makes it sound like they were, but I am uncertain). How were initial adsorption configurations of O and OH chosen--were adsorbates evenly distributed? How were edge sites and different facets accounted for in deciding where to place adsorbates? What ML coverages were considered? [Regarding mixed O-OH phases, there is evidence of that for ORR on Pt surfaces (DOI: 10.1038/ncomms3817) though understandably this is a more complicated system. Would be good to know which design elements were considered in the current manuscript.]

158: "O and OH were adsorbed on either bridge or top sites, which were the most stable adsorption sites on the Pt(111) slab" -> O* usually likes to be on fcc sites of a Pt(111) surface? (<https://doi.org/10.1007/s10562-020-03286-w> and others) Perhaps it is different for NPs since the NPs in the training set are small structures and may not have a nice flat Pt(111) surface. Either way please clarify this and support with a reference or data.

177: "DDE[np - (A_ads)_n]" the np is lowercase and elsewhere it is uppercase NP.

190: "both O and OH species" sounds like mixed phase surface. Please clarify if that is the case, see the comment for line 157.

224: Ih was already defined earlier in line 153, together with Coh.

298: Figure S5, lowest oxygen state for Pt(111) slab is shown as 1ML OH whereas previous literature usually finds 1/3 MH OH-H₂O in a water stabilized case (DOI: 10.1039/b803956a and DOI: 10.1103/PhysRevLett.89.276102). Please include calculation details about this figure: what was the periodic slab size and what surface coverages were considered in constructing the diagram? Was it only 2x2 slabs or 3x3 slabs as well? Were other coverages of OH considered? Perhaps this difference in the S5 figure compared to literature is the lack of an explicit water layer or perhaps due to the sample of coverages considered. Either way would be helpful to clarify that since it currently does not match.

Reviewer #3:

Remarks to the Author:

The use of ML in material engineering is interesting. However, authors must address the following issues:

[1] an introduction to CGCNN with proper citation needed.

[2] a comparative study of CGCNN and the proposed method along with the other traditional method can be more informative.

[3] Architectural details of BE-CGCNN is must. Authors need to provide the detail of the architecture of the proposed graph CNN in the main text.

[4] Dataset details must be included. If possible include the dataset in supplementary.

[5] What is the reason of use mean absolute error as cost function, why not sum of square error?

[6] An intro to Graph CNN and method to use it for material engineering will be more informative.

[7] in total authors need to include substantial content related to ML, GNN, GCNN, CGCNN and proposed BE-CGCNN with substantial architectural details is must.

[8] Authors also need to provide detail of all the alternative dataset for similar work.

[9] Their is lack of information regarding parameters tuning.

Authors must address this issues before potential publication.

Reviewer #4:

Remarks to the Author:

In " Machine Learning-Enabled Exploration of the Electrochemical Stability of Real-Scale Metallic Nanoparticles ", the authors present a number of experiments, which can be of some use if published. However, the manuscript must be revised to rewrite the text, improve the methodology section or scale back the conclusions to appropriate levels to support the claims.

- The study showed that BE-CGCNN can serve as a better tool compared to the conventional DFT and original CGCNN in studying the stability of real-scale and arbitrarily shaped NPs. Nevertheless, the author should list down the limitations of this approach in electrochemical environments.
- The author mentioned that adsorption of OH on a vertex site of Pt55 (Coh) NPs is 0.68 eV stronger than that on a terrace site. The author should show the calculation in detail in the supporting information.
- The author calculated the energy of Pt NPs by the classical forcefield and applied the second nearest-neighbor modified embedded-atom method (2NN MEAM). All the assumptions in the theoretical calculations should be written in details.

Reviewer #1 (Remarks to the Author):

[General review]

The work reports on the machine learnt accelerated DFT-based construction of the Pourbaix Diagram of nanoparticles. The topic is of high relevance both for fundamental and applied science motivations. Further, it demonstrates a step forward in embracing a realistic complexity in modelling, so to close the complexity gap between simulations and experiments.

I'm very positive about the work, and would ask few additional clarifications/calculations, so as to improve the impact and reach of the paper, and fully match the quality of reports disseminated in Nature Communications:

[Our response]

We thank the reviewer for the positive evaluation of our manuscript. We performed additional calculations and analyses to fully comply with the reviewer's comment as below.

[Comment 1]

I think that it is necessary to report on the uncertainty of machine learning models predictors for adsorption energy calculation. This could be readily extracted from, e.g., predictions obtained from committee of models. I refer the authors to Journal of Chemical Physics 154 (7), 074102 2021 for a detailed calculation of unbiased uncertainties given a committee of machine learning models, and to Mach. Learn.: Sci. Technol. 1 025006, 2020 in case they would prefer to adopt alternative uncertainty estimates.

Note, It is also very interesting to assess how uncertainty propagates into the Pourbaix Diagram calculation. I expect that rules of propagation of error would render this rather feasible, and provide an even more grounded estimate of the agreement between the model and experiments.

[Our response]

We thank the reviewer for providing the fruitful comment and relevant reference papers. The uncertainty quantification in ML predictions has become important nowadays so it would be very helpful to estimate the uncertainty of our models. We adopted the approach of Dropout Neural Network (Dropout NN) model from Mach. Learn.: Sci. Technol. 1 025006, 2020. We applied dropout to each convolution layer in BE-CGCNN with the rate of 25%. As a result, we could estimate the uncertainty value for each prediction (**Figure R1a**), and the uncertainty-reflected version of the surface Pourbaix diagram can be constructed, as shown in an example of **Figure R1b**. This model with Dropout NN slightly underperforms than the model without Dropout NN, as confirmed by the increased MAE values of 0.67 eV and 0.19 eV for each O and OH adsorbate, respectively. Despite the reduced accuracy, the qualitative and quantitative trends of the surface Pourbaix diagram are not affected much, as shown in **Figure R1b**.

Figure R1. Uncertainty estimation using Dropout NN and its reflection to surface Pourbaix diagram. **a** ML (BE-CGCNN with Dropout NN) and DFT prediction results of adsorption energy difference. The error bar represents SD values of each point. **b** Surface Pourbaix diagram of Pt₅₅ (Coh) with uncertainty estimations. The phase boundary line is represented by the dashed line (the average of 1,000 sampled cases), and its uncertainty is represented by the gradient at each line. For the clearer understanding of the uncertainty values, the probability density distributions of each phase boundary lines at pH 14 are shown on the right.

[Revision to manuscripts]

We have added **Figure 4** and related explanations in the main text to discuss the uncertainty quantifications in our model, and its effect on the surface Pourbaix diagrams.

(Page 13) “Although we have thus far shown that BE-CGCNN model can produce surface Pourbaix diagrams with fairly high accuracy, it does not tell whether or not the ML predicted results are trustworthy. To estimate the model’s reliability, the uncertainty quantification could be a useful tool. By estimating uncertainty values for ML predictions, the model not only provides a prediction value but also returns confidence intervals. There are many available methods to measure uncertainty^{49, 50, 51}, and among them, we adopted Dropout Neural Network (Dropout NN) which is directly applicable to our BE-CGCNN model with ease. It approximates Bayesian models by enforcing dropout at the ML prediction stages. We used a random dropout rate of 25 % to each convolution layer before fully connected neural network layers in the BE-CGCNN model. 1,000 times of sampling was performed for each prediction, and the uncertainty value can be obtained as a standard deviation of 1,000 times predictions from the committee of ML models.

This model with Dropout NN was trained with each O and OH DB, and prediction result is shown in Figure 4a. It slightly underperforms the model without Dropout NN, as confirmed by the increased MAE values of 0.67 eV and 0.19 eV for each O and OH adsorbates. On the other hand, the SD values are 0.017 eV and 0.015 eV for each O and OH DB, respectively, which are very small values considering the very wide range of adsorption energy difference in our dataset. This result indicates that BE-CGCNN model is highly reliable. In addition, surface Pourbaix diagram can be built based on the model with Dropout NN, as shown in the example of Pt₅₅ (Coh) in Figure 4b. In the diagram, the main phase boundary lines were determined from the average of the ML prediction values (average of 1,000 sampled cases), while the uncertainty of boundary lines were calculated by adding and subtracting of SD values

at each line. We observe in Figure 4b that the Pourbaix diagram of Pt₅₅ (Coh) is qualitatively and quantitatively similar to the results obtained without Dropout NN, since the prediction accuracy is not much undermined. The uncertainty range of phase boundary lines would reveal the reliability of the boundary lines. In the case of Pt₅₅ (Coh) in Figure 4b, the largest standard deviation of phase boundary line is only 0.117 V for the phase boundary between Pt₅₅-(O)_{0.56ML} and Pt₅₅-(O)_{0.67ML}, and the other boundary lines are much more confident as confirmed by smaller standard deviations (0.031~0.066 V). The BE-CGCNN model with Dropout NN is highly beneficial for predicting the uncertainty of the predicted phase boundary lines of Pourbaix diagrams. Nonetheless, since the overall trends of surface Pourbaix diagram are not affected much by the inclusion of Dropout NN, the remaining studies were performed with BE-CGCNN without Dropout NN.”

[Comment 2]

It would be interesting to report on how the Pourbaix Diagram changes depending on the DFT functional choice. While RPBE+D3 is a perfectly reasonable choice to calculate adsorption energies, other properties e.g., thermodynamic stability of a given nanostructure, are better captured by other DFT functionals (see e.g., Nature Communications 12 (1), 1-9 2021, <https://doi.org/10.1038/s41467-021-26199-7>).

[Our response]

We thank the reviewer for raising the comments about DFT functionals. To understand the effect of DFT functionals, we carried out a comparative study using Pt₅₅(Ih)-(OH)_{1ML} with different functionals as follows: LDA, RPBE, and RPBE+D3. We first compared the adsorption energies of Pt₅₅(Ih)-(OH)_{1ML} of these three functionals. The adsorption energy per adsorbate with the RPBE functional was -2.50 eV/adsorbate, which is weaker than the RPBE+D3 functional (-2.85 eV/adsorbate). This is likely because the long-range interaction between OH adsorbates is not well described with RPBE functionals without D3 correction term. In contrast, the adsorption energy with LDA was computed as -3.71 eV/adsorbate, which is much stronger than the value with RPBE+D3. **Fig. R2** shows the surface Pourbaix diagrams of Pt₅₅(Ih) are compared with LDA and RPBE+D3 functional. In the diagram for LDA, the OH- and O-covered phase is observed much larger than the case with RPBE+D3 functional. This difference is mainly because LDA usually overestimates adsorption energy on transition metal surface [doi: 10.1021/acs.jpcc.7b12258, DOI:10.1063/1.1328042]. Hybrid functionals or high-level computation methods such as perturbation theory would calculate adsorption energies more accurately but they would require too much computational costs. Therefore, RPBE+D3 functional would be the most reasonable functional to build adsorption energy dataset. We added the reasons to choose RPBE+D3 functional in Computational Method section.

Figure R2. The comparison of Pourbaix diagrams of Pt₅₅(lh) obtained by (a) LDA functional and (b) RPBE+D3 functional.

[Revision to manuscript]

We discussed the reasons to choose RPBE+D3 functional in Computational Method section, and also added the **Figure S8**.

(Page 19) “Another function of local-density approximation (LDA) was also tested, and its results are compared to the case with RPBE+D3 in Figure S8. LDA was found to generally overestimate adsorption energies, causing unrealistic Pourbaix diagrams. In contrast, RPBE+D3 is known to produce accurate adsorption energies with reasonable computational cost compared to other functionals^{62, 63}.

[Comment 3]

The discussion presented in Adsorption energy dataset for various NP surface coverages should be clarified. In particular “and M is the number of each n adsorbate case in our dataset” is a confusing sentence. yet, it is of key importance to understand the metric introduced by the authors.

[Our response and revision to manuscript]

We thank the reviewer for this comment and modified the description of M in the main text, as below. Also, we provided the M values in the below Table R1 (or **Table S2** in Supplementary Information) to reduce the confusion.

(Page 8) “where $\Delta\Delta E[\text{NP}-(A_{\text{ads}})_n]$ is the adsorption energy difference, $\overline{\Delta E_{\text{ads}}[\text{NP}-(A_{\text{ads}})_n]}$ is an averaged value of adsorption energies per adsorbate at each n, and **M is the number of**

adsorption energy data for each n and each adsorbate, and these M values are provided in Supplementary Table S2.”

NP	Number of adsorption energy data	
	O	OH
Pt ₁₃ (Ih)	48	49
Pt ₁₃ (Coh)	51	53
Pt ₃₈ (Toh)	47	38
Pt ₅₅ (Ih)	104	109
Pt ₅₅ (Coh)	105	110
Pt slab	14	8
Total	369	367

Table R1. Number of adsorption energy data (M) for each n (n as in NP-(A_{ads})_n) and each adsorbate (O or OH).

[Comment 4]

In the abstract the authors highlight the capability of the model for "arbitrarily shaped" NPs, yet they present a discussion solely on closed-shell nanoparticles with no defects. Can the authors demonstrate the effectiveness of the model for defected/asymmetric shapes? are the trends they observe confirmed also for non-idealised shapes?

Note, the authors correctly included less-ideal shape in the training of machine learning models for DOS in nanoparticles (Scientific Reports 11, 11604 (2021)). Also, defected Pt nanoparticles may be very active for, e.g., Oxygen Reduction (see J. Am. Chem. Soc. 2020, 142, 42, 17812–17827 and Acs Catalysis 10 (6), 3911-3920, 2020 just to name a few).

[Our response]

We thank the reviewer for raising this important comment. We performed additional experiments to understand whether or not our model is applicable to non-idealized shapes. Following the reviewer’s comment, we generated two asymmetric NPs (AS1 and AS2, hereafter) using the heating-and-quenching approach in molecular dynamics simulations (LAMMPS program). The procedure for generating AS1 and AS2 NPs is composed of four steps: (1) prepare a NP cluster composed of 55 atoms, (2) For 200 ps, heat the box up to 1000 K which is higher than the melting temperature of the Pt cluster, (3) For 0.1 ps, rapidly cool down the box to 10 K to main the asymmetric shapes, and (4) finally perform DFT relaxations to obtain local minimum structures.

In **Figure R3**, we present the surface Pourbaix diagrams of these AS1 and AS2 NPs, obtained by DFT and BE-CGCNN model. We found that the same phases appear in both diagrams and the trend is very similar, which supports the effectiveness of BE-CGCNN for non-idealized NP shapes.

Figure R3. Surface Pourbaix diagrams of asymmetric NPs (AS1 and AS2) based on DFT calculation or BE-CGCNN prediction.

[Revision to manuscript]

We added the following discussions regarding asymmetric NPs in the main text, as below. And, we added **Figure S6** in Supplementary Information to provide detailed information of the asymmetric NP generation process as well as their Pourbaix diagrams.

(Page 12) “The BE-CGCNN model is not only applicable to highly symmetric NPs such as Ih and Coh, but also to asymmetric NPs which could be also synthesized in experimental condition^{47, 48}. To validate the performance of BE-CGCNN model for the non-idealized NP shapes, surface Pourbaix diagram of asymmetric NPs were built based on the prediction of adsorption energy difference. For more realistic modeling, the shapes of asymmetric NPs were generated using the heating-and-quenching approach of Pt₅₅ NPs in molecular dynamics (MD) simulations. As a result, two different structures of asymmetric Pt₅₅ NP structures (AS1, AS2) could be obtained. The details of the structure generation procedures are provided in Supplementary Figure S6. In Figure S6, the surface Pourbaix diagrams of Pt₅₅(AS1) and Pt₅₅(AS2) are shown and compared with DFT-computed diagrams. Because asymmetric NP structures are not included in training set, the difference between BE-CGCNN and DFT based diagrams are larger than the difference for the case of Pt₅₅(Ih) or Pt₅₅(Coh). Nevertheless, the

same phases appear in both diagrams and the trend is very similar, which proves the effectiveness of BE-CGCNN for non-idealized NP shapes.

Interestingly, the shape of diagrams of asymmetric NPs are quite different from that of Pt₅₅(Ih) despite the same size. The main difference is that the area of bare Pt phase is expanded for AS1 and AS2 NPs. As the asymmetric NP passes through melting and quenching process, the surface become smoothed, and thus the number of low-coordinated surface atoms such as vertex and edge (typically stronger binding sites) become lower. Therefore, the average adsorption energy per adsorbate become weaker from -2.90 eV for OH and -4.18 eV for O on Ih and Coh NPs to -2.35 eV for OH and -3.61 eV for O on AS1 and AS2 NPs. This result adequately explains the expanded bare Pt region in Pourbaix diagrams for asymmetric NPs.”

(In the caption of **Figure S6**) “These asymmetric nanoparticles were generated using the heating-and-quenching approach in molecular dynamics simulations (LAMMPS program). The procedure is composed of four steps: (1) prepare a NP cluster composed of 55 atoms, (2) For 200 ps, heat the box up to 1000 K which is higher than the melting temperature of the Pt cluster, (3) For 0.1 ps, rapidly cool down the box to 10 K to main the asymmetric shapes, and (4) finally perform DFT relaxations to obtain local minimum structures. Note that the MD time-step was chosen as 1 fs, and the canonical (NVT) ensemble was used. The box size for the MD simulation was 35×35×35 Å³, in which one NP structure was included.”

Reviewer #2 (Remarks to the Author):

[General comments]

This manuscript addresses the complexity of predicting surface Pourbaix diagrams for large-size nanoparticles that would ordinarily be outside of the bounds of ab initio DFT. By using an extension to CGCNNs, this machine learning approach accounts for four bonding types explicitly. The reproduction of surface Pourbaix diagrams that were created with DFT shows the utility of this approach and as such would be of impact when considering stability of realistic systems. Primarily the comments included here deal with the reporting of methodological details. I do not think the information currently provided is enough to reproduce results so it would be quite helpful to have more insight into the how the training dataset was defined.

[Our response]

We thank the reviewer for the positive evaluation of our manuscript. We performed additional calculations and analyses to fully address the reviewer's concerns and comments as below.

[Comment 1]

133: The intuition for choosing the feature set seems reasonable, coupled with the figure 1c. However it would be helpful if the cost function were explicitly defined somewhere. The differences in cost seem quite small and I would naively expect that a combination of features that least relate to adsorption energy and valance electrons would result in a higher cost and lower accuracy.

[Our response and revision to manuscript]

We used L^2 loss (mean squared error, MSE) with L^2 regularization as a cost function of BE-CGCNN model because it is known to be converged stably compared to L1 loss function (mean absolute error, MAE). Nevertheless, the model with low cost would also have low MAE and high accuracy, so we could get atom features with high accuracy by just comparing cost functions. As you commented, we described the type of cost function in Computational Methods as below.

(Page 22) “The cost function of model was set to L^2 loss with L^2 regularization.”

[Comment 2]

139: Fig 1b typo in "non-boned interaction u4". Text in figure 1b is a bit inconvenient to read on top of the atoms. Also why is u4 shown in red? I thought it was related to the red text in 1c but that is not the case.

[Our response and revision to manuscript]

We modified the typo and used translucent background to increase the readability. The reason we used the red color for u₄ was to highlight the use of non-bonded interaction in bond-

encoding. To avoid confusion, we changed the color to blue. These changes are presented in **Figure R4**, and reflected in Figure 1 of the main manuscript.

Figure R4. Description of the bond-type embedded CGCNN (BE-CGCNN) model. This revised figure corresponds to **Figure 1** of the main manuscript.

[Comment 3]

157: Would be great to know more details about the kind of adsorption configurations were used in the dataset to construct the CGCNNs. This is a large sample space after all (In 190: "736 adsorption energy difference points). Perhaps partly this information may be added in the SI. For example: How many adsorbates are in 1ML on each NP (and on what sites)? Were mixed phases considered between O and OH adsorbates (line 157 makes it sound like they were, but I am uncertain). How were initial adsorption configurations of O and OH chosen--were adsorbates evenly distributed? How were edge sites and different facets accounted for in deciding where to place adsorbates? What ML coverages were considered? [Regarding mixed O-OH phases, there is evidence of that for ORR on Pt surfaces (DOI: 10.1038/ncomms3817) though understandably this is a more complicated system. Would be good to know which design elements were considered in the current manuscript.

[Our response]

We thank the reviewer for raising this comment. Let me answer your questions one by one, as follows.

1) The number of adsorbates in 1 ML for each NP is provided in below **Table R2**. Therefore, we added this table as **Table S3** in Supplementary Information.

2) We did not consider the O and OH mixed phases, because it could be too complicated as you mentioned. Moreover, the current BE-CGCNN model is trained and tested separately with O- or OH-adsorbed structures so our model cannot be applied to mixed phase. However, since the

adsorption energy of the mixed phase should be in between O- and OH-adsorbed structures so we expect that the mixed phase, if present, would appear in between OH- and O-adsorbed phases.

3) The configurations below 1 ML were chosen by randomly distributing the adsorbates into all possible adsorption sites such as vertex, edge, and terrace. As adsorption energies could be changed with configuration, we modeled up to 5 configurations for the same coverage.

NP	Number of adsorbates in 1ML coverage	
	O	OH
Pt ₁₃ (Ih)	24	12
Pt ₁₃ (Coh)	24	12
Pt ₃₈ (Toh)	44	32
Pt ₅₅ (Ih)	60	42
Pt ₅₅ (Coh)	72	42
Pt ₁₄₇ (Ih)	150	92
Pt ₁₄₇ (Coh)	168	92
Pt ₅₆₁ (Coh)	440	252
Pt ₃₈₇₁ (Coh)	1680	1002
Pt ₆₅₂₅ (Coh)	2400	1442

Table R2. Number of O and OH adsorbates in 1ML coverage

[Revision to manuscript]

We added the below descriptions regarding the adsorption configurations in the manuscript, and also added the **Table S3** in Supplementary Information.

(Page 7) “For each NP and slab structure, adsorption configurations of O and OH up to 1 monolayer (ML) were modeled. Here, 1 ML means that adsorbates are fully covered without interacting between neighbored adsorbates. The number of adsorbates for 1 ML for each structure are provided in Table S3 in Supplementary Information. OH was adsorbed on either bridge or top sites and O was adsorbed on bridge or fcc hollow sites, which were the most stable adsorption sites on the Pt(111) slab. For each coverage below 1 ML, up to 5 random configurations where adsorbates are distributed randomly including vertex, edge, and terrace site were modeled to address the effect of configuration to adsorption energies. Configurations with O adsorption-only or OH adsorption-only were considered to reduce the complexity of dataset.”

[Comment 4]

158: "O and OH were adsorbed on either bridge or top sites, which were the most stable adsorption sites on the Pt(111) slab" -> O* usually likes to be on fcc sites of a Pt(111) surface? (<https://doi.org/10.1007/s10562-020-03286-w> and others) Perhaps it is different for NPs since the NPs in the training set are small structures and may not have a nice flat Pt(111) surface. Either way please clarify this and support with a reference or data.

[Our response and revision to manuscript]

We thank the reviewer for raising this comment. In fact, we modeled OH adsorption on bridge or top sites and O adsorption on bridge or fcc hollow sites. Although the adsorption energy of O on fcc hollow site (-4.293 eV) is stronger than bridge site (-4.284 eV), the difference is quite small so we assume that O could not only adsorb on fcc hollow site but also bridge site: we considered both configurations. We clarified the adsorption site of OH and O separately as below.

(Page 7) “OH was adsorbed on either bridge or top sites and O was adsorbed on bridge or fcc hollow sites”

[Comment 5]

177: "DDE[np - (A_{ads})_n]" the np is lowercase and elsewhere it is uppercase NP.

[Our response and revision to manuscript]

We thank the reviewer for capturing this typo and modified it as below.

(Page 8) “To overcome this limitation, we introduced a different metric, namely, the adsorption energy difference ($\Delta\Delta E[\text{NP} - (A_{\text{ads}})_n]$), which served as a much more suitable form for accurate ML training and prediction and could be computed as follows:”

[Comment 6]

190: "both O and OH species" sounds like mixed phase surface. Please clarify if that is the case, see the comment for line 157.

[Our response and revision to manuscript]

As we replied earlier in **Comment 3**, we did not consider the mixed phases. We clarified the expressions as below.

(Page 8) “In Fig. 2, we show the BE-CGCNN model training results obtained using the dataset comprising 736 adsorption energy difference data points (both O adsorption case and OH adsorption case), in which 80% of the dataset was used for training and the remaining 20% was used for the test.”

(Page 9) “For each O adsorption and OH adsorption on Pt NPs, the original CGCNN without bond-type differentiation exhibits the worst accuracy.”

[Comment 7]

224: Ih was already defined earlier in line 153, together with Coh.

[Our response and revision to manuscript]

We thank you for the comment and deleted the duplicate expressions.

(Page 10) “The first validation process is to construct reliable Pourbaix diagrams of Pt₅₅(Ih) NPs using the BE-CGCNN model trained on the data of smaller systems, including slabs and smaller NPs (Pt₁₃ and Pt₃₈).”

[Comment 8]

298: Figure S5, lowest oxygen state for Pt(111) slab is shown as 1ML OH whereas previous literature usually finds 1/3 MH OH-H₂O in a water stabilized case (DOI: 10.1039/b803956a and DOI: 10.1103/PhysRevLett.89.276102). Please include calculation details about this figure: what was the periodic slab size and what surface coverages were considered in constructing the diagram? Was it only 2x2 slabs or 3x3 slabs as well? Were other coverages of OH considered? Perhaps this difference in the S5 figure compared to literature is the lack of an explicit water layer or perhaps due to the sample of coverages considered. Either way would be helpful to clarify that since it currently does not match.

[Our response]

To build a surface Pourbaix diagram of Pt(111) slab, we considered 1×1 , $\sqrt{3} \times \sqrt{3}$, and 2×2 surface unit cells. The adsorption on 1×1 surface unit cell was considered as 1ML. The reason for the difference was because we didn't consider explicit water layers as you mentioned. Because water stabilizes the adsorbed OH, OH adsorption with 1 ML was more stable without water. Because it was too complicated to consider explicit water molecules in NP-adsorbate structures, we did not account waters in adsorption on the slab for consistency.

[Revision to manuscript]

We added the detail of surface calculation in the caption of **Figure S5**.

(In the caption of **Figure S5**) “Figure S5. Pourbaix diagram of Pt(111) slab constructed by DFT calculation. For the slab calculation, 1×1 , $\sqrt{3} \times \sqrt{3}$, and 2×2 surface unit cell was considered, and adsorption on 1×1 surface unit cell was considered as 1ML.”

Reviewer #3 (Remarks to the Author):

[General comment]

The use of ML in material engineering is interesting. However, authors must address the following issues:

[Our response]

We thank the reviewer for the positive evaluation of our manuscript. We performed additional calculations and analyses and modified the manuscript to fully address the raised concerns, as below.

[Comment 1]

[1] an introduction to CGCNN with proper citation needed.

[Our response and revision to manuscript]

We thank the reviewer for this comment and added the descriptions regarding why we adopted CGCNN and referenced the paper from Xie and Grossman.

(Page 3) “To overcome this problem, machine learning (ML) is a useful tool. After building the database, training and prediction could be done with personal computers and the computation time is much faster than those of quantum calculations. Not only a speed, but also an accuracy can be achieved with substantial amount of training set.^{27, 28} Many ML frameworks are used in various materials science fields to predict material’s properties from given structures, such as Random Forest Regression^{29, 30, 31}, Gaussian Process Regression³⁰, and XGBoost Regression³². Among them, Crystal Graph Convolutional Neural Network (CGCNN)³³ has many advantages. At first, it can be applied to any kind of material structures by constructing a graph from atomic coordinates, even to NP structures.³⁴ Moreover, by convolution procedure of graph generated from atomic structures, it takes account local atomic interaction between neighbored atoms which directly influence property of materials.”

[Comment 2]

[2] a comparative study of CGCNN and the proposed method along with the other traditional method can be more informative.

[Our response]

In **Figure R5**, we compared ML, GNN, CGCNN (similar to GCNN), and BE-CGCNN with our dataset. We chose ML as a simple multilayer perceptron (MLP) regression. Because our dataset is composed of complicated NP-adsorbate structures, it could extract features from not only small NPs but also large NPs with larger than 6000 atoms. Therefore, the feature selection

was limited to simple values such as the type of NPs and coverage of adsorbates. Because this kind of feature doesn't include sufficient information on atomic structures, the ML model predicted a single value regardless of input structures. GNN was produced by simply replacing the convolution layer of the CGCNN model. Without convolution, the model cannot capture local atomic structures, so the performance was very low. CGCNN is corresponding to the 'without bond type embedding' model in Table 1. As mentioned in manuscript, the performance of BE-CGCNN was highest because it captures important bond information. This result proves the strength of BE-CGCNN compared to other models.

Figure R5 O Adsorption energy difference prediction result using (a) MLP regression, (b) GNN, (c) CGCNN, and (d) BE-CGCNN.

[Revision to manuscript]

We added **Figure S11** in Supplementary Information to provide the comparative study of MLP, GNN, CGCNN, and BE-CGCNN models.

[Comment 3]

[3] Architectural details of BE-CGCNN is must. Authors need to provide the detail of the architecture of the proposed graph CNN in the main text.

[Our response and revision to manuscript]

We thank the reviewer for this comment. We provide the architectural details in Fig. 1a and also the ‘BE-CGCNN development’ paragraph in Computational Methods section. In addition, we provide the below schematic diagram of BE-CGCNN as **Figure S7** in Supplementary Information.

Figure R6. Detailed schematic diagram of BE-CGCNN model.

[Comment 4]

[4] Dataset details must be included. If possible include the dataset in supplementary.

[Our response]

We now provide the details of the dataset in ‘Adsorption energy dataset for various NP surface coverages’ paragraphs in the Results section. We found that some key information to reproduce our results are missing. For example, the number of adsorbates in 1 ML for each NP is provided in below **Table R2**. Therefore, we added this table as **Table S3** in Supplementary Information. In addition, the configurations below 1 ML were chosen by randomly distributing the adsorbates into all possible adsorption sites such as vertex, edge, and terrace. As adsorption energies could be changed with configuration, we modeled up to 5 configurations for the same coverage.

NP	Number of adsorbates in 1ML coverage	
	O	OH
Pt ₁₃ (Ih)	24	12
Pt ₁₃ (Coh)	24	12
Pt ₃₈ (Toh)	44	32
Pt ₅₅ (Ih)	60	42
Pt ₅₅ (Coh)	72	42
Pt ₁₄₇ (Ih)	150	92
Pt ₁₄₇ (Coh)	168	92
Pt ₅₆₁ (Coh)	440	252
Pt ₃₈₇₁ (Coh)	1680	1002
Pt ₆₅₂₅ (Coh)	2400	1442

Table R2. Number of O and OH adsorbates in 1ML coverage

[Revision to manuscript]

We added the below descriptions regarding the adsorption configurations in the manuscript, and also added the **Table S3** in Supplementary Information.

(Page 7) “For each NP and slab structure, adsorption configurations of O and OH up to 1 monolayer (ML) were modeled. Here, 1 ML means that adsorbates are fully covered without interacting between neighbored adsorbates. The number of adsorbates for 1 ML for each structure are provided in Table S3 in Supplementary Information. OH was adsorbed on either bridge or top sites and O was adsorbed on bridge or fcc hollow sites, which were the most stable adsorption sites on the Pt(111) slab. For each coverage below 1 ML, up to 5 random configurations where adsorbates are distributed randomly including vertex, edge, and terrace site were modeled to address the effect of configuration to adsorption energies. Configurations with O adsorption-only or OH adsorption-only were considered to reduce the complexity of dataset.”

[Comment 5]

[5] What is the reason of use mean absolute error as cost function, why not sum of square error?

[Our response and revision to manuscript]

We used MSE with L^2 regularization as a cost function. We described the type of cost function in Computational Methods as below.

(Page 22) “The cost function of model was set to L^2 loss with L^2 regularization.”

[Comment 6]

[6] An intro to Graph CNN and method to use it for material engineering will be more informative.

[Our response and revision to manuscript]

We thank the reviewer for this comment. We used CGCNN based ML model because it can be applied to any kind of atomic structures and take account the interaction between local atomic structures. We described the reason to choose CGCNN in introduction as follows.

(Page 3) “To overcome this problem, machine learning (ML) is a useful tool. After building the database, training and prediction could be done with personal computer and computation time is much faster than quantum calculation. Not only a speed, but also an accuracy can be achieved with substantial amount of training set.^{27, 28} Many ML frameworks are used in materials science field to predict material’s properties from given structures, such as Random Forest Regression^{29, 30, 31}, Gaussian Process Regression³⁰, and XGBoost Regression³². Among them, Crystal Graph Convolutional Neural Network (CGCNN)³³ has many advantages. At first, it can be applied to any kind of material structures by constructing a graph from atomic coordinates, even to NP structures.³⁴ Moreover, by convolution procedure of graph generated from atomic structures, it takes account local atomic interaction between neighbored atoms which directly influence property of materials.”

[Comment 7]

[7] in total authors need to include substantial content related to ML, GNN, GCNN, CGCNN and proposed BE-CGCNN with substantial architectural details is must.

[Our response]

We thank the reviewer for this comment. Following the reviewer’s suggestion, we additionally performed a comparative study of ML (multilayer perceptrons), GNN, CGCNN, and BE-CGCNN models. Please find the results in our responses to **Comment 2**.

[Comment 8]

[8] Authors also need to provide detail of all the alternative dataset for similar work.

[Our response and revision to manuscript]

To build a surface Pourbaix diagram by ML, a dataset of adsorption energy is needed. There are several databases containing adsorption energy data in the materials science and chemistry field such as OC2020 [doi: 10.1021/acscatal.0c04525] and OC2022 [arXiv preprint arXiv:2206.08917]. They include structures of different adsorbates adsorbed on various kinds of material slabs with corresponding adsorption energy data. However, they did not consider enough the coverage of adsorbates and structures of NP-adsorbates, which are critical for

Pourbaix diagram constructions. Therefore, the alternative database would be insufficient to build a surface Pourbaix diagram so we developed our own database by DFT calculation. We added the below descriptions discussing the need of database development by DFT calculation in the manuscript.

(Page 6) “In constructing surface Pourbaix diagram, a dataset of adsorption energies on catalyst surface is required. There are several databases containing adsorption energy data in materials science and chemistry field such as OC2020⁴² and OC2022⁴³, but they did not consider enough the coverage of adsorbates and structures of NP-adsorbates, which are critical for Pourbaix diagram constructions. Therefore, we developed our own database by DFT calculation.”

[Comment 9]

[9] There is lack of information regarding parameters tuning.

[Our response and revision to manuscript]

Four hyperparameters including the number of hidden layers, number of convolution layers, batch size, and learning rate, were fitted by the random search method. The results of hyperparameter fitting are shown in the below **Figure R7**. The result of hyperparameter fitting is added to Supplementary Information as **Figure S9**.

Figure R7 The result of hyperparameter fitting of (a) number of hidden layers, (b) number of convolution layers, (c) batch size, and (d) learning rate.

Reviewer #4 (Remarks to the Author):

[General comment]

In " Machine Learning-Enabled Exploration of the Electrochemical Stability of Real-Scale Metallic Nanoparticles ", the authors present a number of experiments, which can be of some use if published. However, the manuscript must be revised to rewrite the text, improve the methodology section or scale back the conclusions to appropriate levels to support the claims.

[Our response]

We thank the reviewer for the positive evaluation of our manuscript. We performed additional calculations and analyses and modified the manuscript to fully address the raised concerns, as below.

[Comment 1]

The study showed that BE-CGCNN can serve as a better tool compared to the conventional DFT and original CGCNN in studying the stability of real-scale and arbitrarily shaped NPs. Nevertheless, the author should list down the limitations of this approach in electrochemical environments.

[Our response and revision to manuscript]

We thank the reviewer for raising this comment. The main limitation would be that the current model is not universal but dependent on training data. Because our training set is limited to Pt-O and Pt-OH systems, it cannot predict structures of other adsorbates (e.g., OOH, CO) or different compositions of NPs (e.g., Pt₃Ni, Pt₃Fe). However, if we precisely prepare a training dataset for the system we are interested in and train the model carefully, the model would be effective for the expanded material spaces. We added the discussion about the limitation of our model to the manuscript.

(Page 19) "Currently, our model is limited to specific systems as our data set is only composed of Pt-O or Pt-OH structures. The model will not function well for other adsorbates (e.g., OOH, CO) or for different composition of NPs (e.g., Pt₃Ni, Pt₃Fe). However, if we precisely prepare a training set for the system we interested in and follow the similar protocol, BE-CGCNN model would be effective for the expanded material spaces, which remains as a future research."

[Comment 2]

The author mentioned that adsorption of OH on a vertex site of Pt₅₅ (Coh) NPs is 0.68 eV stronger than that on a terrace site. The author should show the calculation in detail in the supporting information.

[Our response and revision to manuscript]

We provided the calculation detail in the ‘DFT computation’ paragraph in ‘Computational Methods’ section. We used VASP with RPBE+D3 functional. The relaxed structure OH adsorption on vertex and terrace sites is shown in **Figure R8**. The vertex site is the corner where two (100) planes and two (111) planes met, and the terrace site is on the (100) plane. The structures of OH adsorbed Pt₅₅(Coh) are added to Supplementary Information as **Figure S10**.

Figure R8. OH adsorbed structure on (a) vertex and (b) terrace site of Pt₅₅ (Coh). ΔE_{OH} is adsorption energy of OH, which is calculated as follows: $\Delta E_{\text{OH}} = E(\text{Pt}_{55}\text{-OH}) - E(\text{Pt}_{55}) - E(\text{OH})$, where $E(\text{Pt}_{55}\text{-OH})$, $E(\text{Pt}_{55})$, and $E(\text{OH})$ are the energies of OH-adsorbed Pt₅₅(Coh), bare Pt₅₅(Coh), and OH radical, respectively.

[Comment 3]

The author calculated the energy of Pt NPs by the classical forcefield and applied the second nearest-neighbor modified embedded-atom method (2NN MEAM). All the assumptions in the theoretical calculations should be written in details.

[Our response and revision to manuscript]

We thank the reviewer for raising this comment. We assumed that the trend of energy of Pt NPs would be not so different between DFT and classical forcefield. As shown in **Figure S3**, the formation energy by 2NN-MEAM follows a similar trend to the DFT calculation. Because the relative energy, not the absolute energy, between NPs is important to calculate the dissolution phase, 2NN-MEAM could be a substitute to DFT. Also, in the evaluation of the forcefield of Pt, MEAM showed quite good performance even for NP structures. [DOI:10.1021/acs.jctc.1c00434]. We added the discussion of the reason to use 2NN-MEAM in the manuscript.

(Page 21) “To overcome this problem, we calculated the energy of Pt NPs by the classical

forcefield. Here, we applied the second nearest-neighbor modified embedded-atom method (2NN MEAM) of J.-S. Kim *et al.*⁷⁰. As shown in Supplementary Figure 3, the energies calculated by the 2NN-MEAM forcefield are approximately 0.3-0.8 eV larger than the DFT calculated energies; however, the trends appear to be very similar. **Because the relative energy, not the absolute energy, between NPs is important to calculate the dissolution phase, 2NN-MEAM could be a substitute to DFT. Also, in the work of evaluation of forcefield for Pt⁷¹, MEAM showed quite good performance even for NP structures.** Thus, we used the 2NN-MEAM forcefield energy for the computation of the Gibbs free energy of the dissolution phase.”

Reviewers' Comments:

Reviewer #1:

Remarks to the Author:

The Authors accurately and coherently addressed the points raised during the first stage of review.

Reviewer #2:

Remarks to the Author:

I find that the reviewer comments were addressed in detail and I generally look favorably on the impact of this manuscript.

Reviewer #3:

Remarks to the Author:

[1] In the BE-CGCNN architecture only one Convolution layer being used. Any reason for that?

[2] The Number of samples seems very small. How to make sure that the network is not overfitted?

Reviewer #4:

Remarks to the Author:

The manuscript "Machine Learning-Enabled Exploration of the Electrochemical Stability of Real-Scale Metallic Nanoparticles" demonstrates the construction of surface Pourbaix diagrams for large-size nanoparticles that would enable the further research on electrochemical stability for various nanoparticle sizes and shapes. The use of machine learning is a novel approach that has the potential to advance the field. Furthermore, the authors have modified the manuscript based on the given the comments. I will recommend this study to be accepted for publication.

Reviewer #3 (Remarks to the Author):

[Comment 1]

In the BE-CGCNN architecture only one Convolution layer being used. Any reason for that?

[Our response and revision to manuscript]

We thank the reviewer for raising this comment. We utilized 5 convolution layers for BE-CGCNN model based on the result of the hyperparameter fitting in **figure R1b**, as we described in Computational Methods section and **figure S7**. However, we acknowledge that the schematic diagram of the model could be misleading as it suggested the presence of only one convolution layer. To address this issue, we updated the schematic diagram by replacing the term ‘Convolution’ with ‘Convolution \times N times’, as shown in **figure R2a**. The change was reflected to **figure 1**.

Figure R1 The result of hyperparameter fitting of (a) number of hidden layers, (b) number of convolution layers, (c) batch size, and (d) learning rate. Red box indicates the selected parameter values. This figure corresponds to **figure S9** in Supplementary.

Figure R2. Description of the bond-type embedded CGCNN (BE-CGCNN) model. This revised figure corresponds to **figure 1** of the main manuscript.

[Comment 2]

The Number of samples seems very small. How to make sure that the network is not overfitted?

[Our response]

We thank the reviewer for raising this comment. We have used Dropout and L^2 regularization to reduce the overfitting. Moreover, to check whether the model is overfitted or not, we trained the model for 1,000 epochs and compared the cost between training and validation set as shown in figure R2. To avoid the overfitting, the parameters where the cost of validation set is the lowest were selected. We described the process to avoid the overfitting in Computational Methods section and the learning curve was added in Supplementary as **figure S12**.

Figure R3. Cost of training and validation set at every 5 steps during training process of (a) O adsorption DB and (b) OH adsorption DB. The minimum value of the cost of validation set is denoted as a blue star. This figure corresponds to **figure S12** in Supplementary.

(page 22) “We trained the model for 1,000 epochs and the parameter where the cost of validation set is the lowest were selected, as shown in figure S12.”

Reviewers' Comments:

Reviewer #3:

Remarks to the Author:

Authors have addressed the concerns. The manuscript may be considered for publication.